# A Bionic-Based Multi-Objective Optimization for a Compact HVAC System with Integrated Air Conditioning, Purification, and Humidification

**DOI:** 10.3390/biomimetics10030159

**Published:** 2025-03-03

**Authors:** He Li, Bozhi Yang, Xinyu Gu, Wen Xu, Xuan Liu

**Affiliations:** 1School of Sciences, Changzhou Institute of Technology, Changzhou 213032, China; 2School of Mathematics and Statistics, Anyang Normal University, Anyang 455000, China

**Keywords:** multifunctional device, biomimetic design, multi-objective optimization, slime mold algorithm (SMA), deep ACO algorithm, beluga whale optimization (BWO)

## Abstract

This study is dedicated to the development of a multifunctional device that integrates air conditioning, humidification, and air purification functions, aimed at meeting the demands for energy efficiency, space-saving, and comfortable indoor environments in modern residential and commercial settings. The research focuses on achieving a balance between performance, energy consumption, and noise levels by combining bionic design principles with advanced optimization algorithms to propose innovative design and optimization methods. Specific methods include the establishment and optimization of mathematical models for air conditioning, air purification, and humidification functions. The air conditioning module employs a nonlinear programming model optimized through the Parrot Optimizer (PO) Algorithm to achieve uniform temperature distribution and minimal energy consumption. The air purification function is based on a bionic model and optimized using the Deep ACO Algorithm to ensure high efficiency and low noise levels. The humidification function utilizes a mist diffusion model optimized through the Slime Mold Algorithm (SMA) to enhance performance. Ultimately, a multi-objective optimization model is constructed using the Beluga Whale Optimization (BWO), successfully integrating the three main functions and designing a compact segmented cylindrical device that achieves a balance of high efficiency and multifunctionality. The optimization results indicate that the device exhibits superior performance, with a Clean Air Delivery Rate (CADR) of 400 m^3^/h, a humidification rate of 1.2 kg/h, a temperature uniformity index of 0.08, and a total power consumption controlled within 1600 W. This study demonstrates the significant potential of bionic design and optimization technology in the development of multifunctional indoor environment control devices, enhancing not only the overall performance of the device but also the comfort and sustainability of the indoor environment. Future work will focus on system scalability, experimental validation, and further optimization of bionic characteristics to expand the device’s applicability and enhance its environmental adaptability.

## 1. Introduction

Heating, ventilation, and air conditioning (HVAC) systems are essential components of modern buildings, accounting for approximately 40% of total energy consumption in the built environment [1]. These systems play a critical role in maintaining indoor thermal comfort, regulating air quality, and managing humidity levels. Over the years, extensive research efforts have been directed at improving the energy efficiency of HVAC systems and adapting them to address increasing concerns over indoor air quality (IAQ), thermal comfort, and sustainability [2]. In urban environments, rising levels of air pollution from contaminants such as nitrogen dioxide (NO_2_) and particulate matter have further necessitated advancements in air purification technologies [3]. As urbanization continues to expand and energy consumption becomes a central concern in sustainable development, enhancing the performance and efficiency of HVAC systems has become an urgent priority.

Despite significant progress in the development of individual HVAC components, traditional systems are often designed as fragmented units, prioritizing specific functions—such as cooling, heating, or humidity control—without achieving holistic optimization. This fragmentation leads to inefficiencies, including suboptimal energy performance, system redundancies, and challenges in balancing energy efficiency with occupant comfort. For instance, a lack of integration between temperature regulation, humidity management, and air purification functions prevents the effective optimization of IAQ, resulting in disjointed performance [4,5]. Moreover, standalone systems frequently operate without coordination, contributing to excessive energy consumption and increased operational costs, which are both economically and environmentally unsustainable [6]. Additionally, achieving a balance between energy efficiency, noise reduction, and occupant comfort remains a persistent challenge, particularly in densely populated urban settings where energy demand is high and environmental quality is a growing concern [7,8].

To address these challenges, the integration of temperature regulation, humidity control, and air purification into multifunctional HVAC devices has emerged as a promising solution. Such integrated systems offer the potential to significantly enhance energy efficiency, improve system functionality, and provide a more comprehensive approach to occupant comfort and environmental sustainability. However, current multifunctional systems face limitations in their design, such as suboptimal performance trade-offs and the lack of advanced frameworks to manage the interactions between their components. These limitations highlight the need for innovative strategies that not only optimize individual functions but also ensure seamless integration and operational synergy.

This study seeks to address these challenges by providing a comprehensive review of recent advancements in HVAC technologies, including innovations in passive cooling techniques, air purification methods, and advanced humidity control systems. Furthermore, the limitations of current multifunctional systems are critically examined to identify gaps in existing research. Building on these insights, the study proposes strategies to guide the development of next-generation HVAC systems. These include leveraging bionic design principles, implementing advanced optimization algorithms, and developing empirical validation frameworks to enhance performance, energy efficiency, and sustainability. By bridging these research gaps, this work aims to contribute to the design of HVAC systems that align with the demands of modern, sustainable buildings while addressing the complexities of urban environments. Through this approach, the study aspires to promote the evolution of HVAC systems from fragmented and energy-intensive units to integrated and efficient solutions capable of meeting the requirements of future urban and residential environments.

## 2. Literature Review

### 2.1. Historical Development of HVAC Systems

#### 2.1.1. Energy Efficiency and Passive Cooling Techniques

The initial focus of HVAC research was on improving energy efficiency. Clarke [1] introduced energy simulation techniques for optimizing HVAC systems and integrating renewable energy technologies into building designs. This work laid the foundation for energy-efficient HVAC systems, enabling better energy modeling and control in buildings.

Pfafferott et al. [4] expanded on this by examining passive cooling strategies, such as night ventilation, which reduce reliance on mechanical cooling. Their parametric modeling and simulation-based studies demonstrated the potential of passive techniques in enhancing thermal comfort and minimizing energy use in hot climates. Building on these principles, Alemu et al. [5], who proposed the energy-saving optimization of HVAC systems using the Ant Lion Optimizer (ALO), achieving significant improvements in energy efficiency.

#### 2.1.2. Machine Learning and Computational Optimization

The integration of computational methods into HVAC optimization represented a paradigm shift. Kusiak and Xu [9] utilized dynamic neural networks to optimize HVAC system performance, showcasing the potential of machine learning in reducing energy consumption. Singh and Abbassi [8] extended this work by combining computational fluid dynamics (CFD) with artificial neural networks (ANNs) to model transient thermal behavior in off-highway machinery cabins. Their approach highlighted the role of hybrid computational techniques in improving thermal comfort and system efficiency. Recently, Cui and Liu [7] introduced a hierarchical control method combining soft Actor–Critic algorithms and hybrid search optimization, achieving significant energy savings while enhancing IAQ in complex building environments.

### 2.2. Air Purification Technologies

#### 2.2.1. Development of Filtration and Photocatalytic Systems

Indoor air quality (IAQ) emerged as a critical focus in HVAC research in the 1990s. Ekberg [2] investigated the impact of outdoor NO_2_ concentrations on IAQ, emphasizing the need for effective air filtration solutions. Over the years, significant advancements have been made, including the development of antimicrobial nanoparticle-coated electrostatic air filters, which achieve high filtration efficiency with low pressure drop [10].

Another promising technology is photocatalytic oxidation (PCO), which removes VOCs and airborne pollutants through catalytic reactions under ultraviolet light. Zhong et al. [11] developed a model to predict PCO efficiency in indoor applications, providing a foundation for understanding its physical and chemical mechanisms. Later, Zhong and Haghighat [12] reviewed PCO materials and technologies, identifying limitations such as byproduct formation and challenges in scalability.

Despite the effectiveness of these technologies, recent advancements have focused on addressing their limitations and improving their integration with HVAC systems. For instance, Pengantis et al. [13] introduced humidity-aware model predictive control for residential HVAC systems, highlighting the potential for real-time air quality optimization. This approach integrates advanced control strategies with air purification systems, ensuring that indoor air quality is maintained efficiently while minimizing energy consumption.

Additionally, Lau et al. [14] demonstrated the use of hollow-fiber membrane technologies in filtration systems, achieving precise air quality control with significantly reduced energy requirements. These innovations not only enhance filtration efficiency but also pave the way for the development of multifunctional systems that combine air purification with other HVAC functions, such as temperature and humidity regulation, into a cohesive and efficient unit.

These recent advancements underscore the importance of combining cutting-edge technologies with intelligent optimization strategies to overcome traditional barriers in air purification systems, particularly in achieving seamless integration with modern HVAC solutions. Future research should focus on scaling these technologies for broader application and addressing challenges such as maintenance requirements and long-term durability in dynamic operational environments.

#### 2.2.2. Integration Challenges

Despite their effectiveness, air purification systems are often designed as standalone units, limiting their integration into HVAC systems. This design limitation poses challenges for achieving seamless functionality and maximizing system efficiency. Haghighat Mamaghani et al. [15] emphasized the critical need for developing more durable and efficient photocatalytic oxidation (PCO) materials that can be easily incorporated into multifunctional systems. Similarly, Izadyar and Miller [16] reviewed the impacts of ventilation strategies and design considerations on indoor airborne transmission, highlighting the importance of integrating air purification systems within HVAC frameworks to enhance both air quality and system performance. These studies collectively underscore the necessity of innovative designs and materials that enable the smooth integration of purification technologies into existing HVAC infrastructures.

### 2.3. Advances in Humidification Technologies

Humidity control plays a crucial role in maintaining IAQ and occupant comfort, particularly in dry or tropical climates. Subramanyam et al. [17] explored the use of desiccant wheels for humidity regulation, demonstrating their potential to improve HVAC system efficiency. Building on this, Wang and Xia [18] optimized desiccant wheel cooling systems, balancing temperature and humidity control to achieve better thermal comfort.

#### Recent Innovations

Recent innovations in humidification technologies have focused on enhancing energy efficiency, improving performance, and enabling sustainable solutions for HVAC systems. These advancements include the following:(a)Liquid dehumidification/humidification cycles: Cai et al. [19] developed a heat pump-based system that integrates liquid dehumidification and humidification, achieving simultaneous temperature and humidity regulation. Additionally, Salins et al. [20] assessed a sustainable technology-based novel humidifier system, which emphasizes energy-efficient performance while maintaining environmental friendliness, further highlighting the potential of integrating innovative approaches into HVAC design.(b)Hollow-fiber membrane humidifiers: Jo et al. [21] demonstrated the effectiveness of hollow-fiber membrane humidifiers in delivering precise humidity control with low energy consumption, making it a promising solution for advanced HVAC applications. This approach aligns with the findings of Solomakha et al. [22], who conducted experimental investigations into the mass transfer characteristics of centrifugal humidifiers, providing insights into optimizing humidity control efficiency.(c)Microwave-based sterilization in humidification systems: Lv et al. [23] explored the application of microwave sterilization technology in central air conditioning systems, showcasing its ability to effectively sterilize humidifiers while maintaining performance. This innovation adds a critical layer of hygiene to HVAC operations, further contributing to the development of safe and sustainable humidification technologies.

These advancements represent significant strides in the development of energy-efficient and sustainable humidification technologies, addressing both operational performance and environmental concerns.

### 2.4. Multifunctional HVAC Systems

The demand for compact and energy-efficient devices has driven interest in integrating multiple HVAC functions into a single system. Dong et al. [24] studied a multifunctional air conditioner integrated with a water heater, showing significant energy efficiency gains. Similarly, Nada et al. [6] proposed a hybrid air conditioning system that combined humidification, dehumidification, and air conditioning, achieving energy savings and water production in arid regions.

He et al. [25] investigated the use of surplus air thermal energy from HVAC systems to heat solar greenhouses, demonstrating an innovative application of thermal energy reuse. These studies highlight the growing trend of multifunctional HVAC systems, but significant challenges remain in optimizing system integration, performance, and cost.

Recent studies have further expanded on optimizing multifunctional HVAC systems through advanced control strategies and energy-efficient designs. Yang et al. [3] developed a distributed control approach for multizone HVAC systems, taking into account indoor air quality and thermal comfort. Their method addresses the complexities of controlling HVAC systems in environments with varying air quality requirements. Furthermore, Y. P. et al. [26] proposed an energy consumption optimization framework for HVAC systems based on deep reinforcement learning. This approach achieves real-time energy savings while ensuring operational efficiency. Similarly, Wang et al. [27] introduced a deep reinforcement learning-based optimization model for HVAC systems in multi-variable air volume (VAV) open offices, demonstrating significant improvements in energy efficiency and occupant comfort.

These studies highlight the growing trend of leveraging advanced technologies such as artificial intelligence and distributed control methods to optimize the performance of multifunctional HVAC systems. However, challenges remain in balancing system integration, cost, and operational performance, which require further exploration in future research.

### 2.5. Optimization and Bionic Design

#### 2.5.1. Optimization Techniques

Optimization algorithms play a pivotal role in enhancing the performance, energy efficiency, and overall reliability of multifunctional HVAC systems. By enabling systems to effectively balance competing objectives—such as minimizing energy consumption, maximizing occupant comfort, and improving operational responsiveness—optimization methods have become essential tools in the design and control of advanced HVAC solutions.

Recent research has introduced a variety of innovative algorithms tailored to address the complex, multi-objective nature of HVAC optimization. Abdoalnasir et al. [28] demonstrated the potential of the Big-Bang Big-Crunch (BB-BC) controller to enhance system responsiveness and energy efficiency, particularly by integrating fractional-order control mechanisms. Similarly, Cen et al. [29] developed a multi-strategy sparrow search algorithm (SSA), which effectively combines diverse search strategies to accelerate convergence and improve optimization performance in central HVAC systems. These advancements underscore the increasing reliance on heuristic and hybrid methods to tackle the inherent complexities of HVAC systems.

Emerging optimization techniques, inspired by biological and natural phenomena, have further expanded the capabilities of HVAC optimization. The Parrot Optimizer (PO), introduced by Lian et al. [30], leverages the intelligent problem-solving behavior of parrots, employing robust global search strategies to navigate multi-dimensional and multi-objective challenges. Similarly, the Slime Mold Algorithm (SMA), proposed by Li et al. [31], mimics the adaptive foraging behavior of slime molds, making it particularly effective for solving stochastic optimization problems and balancing exploration with exploitation in uncertain scenarios.

In addition to these bio-inspired methods, hybrid algorithms have gained significant traction. The DeepACO Algorithm, presented by Ye et al. [32], integrates neural networks into the Ant Colony Optimization framework, enabling the system to adaptively learn and optimize solutions for high-dimensional, combinatorial problems. This neural-enhanced approach has shown significant promise in handling dynamic HVAC control scenarios. Furthermore, the Beluga Whale Optimization (BWO) algorithm, developed by Huang and Hu [33], incorporates multi-strategy mechanisms inspired by beluga whale behaviors, delivering robust global optimization capabilities. This method has proven effective in solving multi-variable and system-level engineering problems, including the design and operation of HVAC systems.

Together, these advanced optimization algorithms provide innovative solutions for addressing the challenges faced by multifunctional HVAC systems. Their ability to handle dynamic, high-dimensional, and stochastic problems makes them invaluable for enhancing energy efficiency, improving indoor air quality, and optimizing system performance. As the complexity of HVAC systems continues to grow, these algorithms represent critical tools for driving the development of sustainable and intelligent environmental control solutions.

#### 2.5.2. Bionic Design Principles

Nature-inspired designs have shown promise in improving system performance. Bixler and Bhushan [34] applied shark skin-inspired microstructures to reduce drag and enhance airflow efficiency, while Gatica et al. [35] demonstrated the use of honeycomb structures for adsorption in water purification systems. These designs optimize airflow paths, reduce energy consumption, and improve overall device functionality.

### 2.6. Summary of Combined Techniques

To enhance the clarity and ease of comparison between the various combined techniques discussed in this section, we present a summary of their respective advantages and disadvantages in Table 1. This table highlights the key strengths and limitations of each approach, providing a clearer understanding of their applicability in different HVAC system designs. By examining these techniques side-by-side, readers can better appreciate the trade-offs involved in integrating multiple functions into advanced HVAC systems.

### 2.7. Research Gaps and Future Directions

#### 2.7.1. Identified Research Gaps


(a)Integration Limitations: Current systems lack a unified approach to integrating air conditioning, humidification, and air purification [6,15](b)Performance Trade-offs: Existing models often require compromises between energy efficiency, IAQ, and noise reduction [7,29](c)Data Deficiency: The absence of high-resolution real-world performance data limits model accuracy and applicability [15].


#### 2.7.2. Future Directions


(a)Develop integrated optimization frameworks for multi-objective HVAC systems.(b)Incorporate bionic designs to enhance airflow dynamics and reduce operational costs.(c)Validate models with real-world datasets to ensure practical applicability and scalability.


## 3. Methodology

### 3.1. Air Conditioning Partial Modeling

#### 3.1.1. Thermal Distribution Within the Conditioned Space

This study is a numerical simulation aimed at validating the model’s effectiveness through optimization algorithms. Experimental validation will be conducted in future work. To achieve uniform temperature distribution and energy efficiency in a room of (*V_room_* = 5 m × 8 m × 3 m = 120 m^3^), a detailed analysis of heat transfer dynamics is essential. The thermal behavior of the room is influenced by both conductive and convective processes.

Heat conduction dominates the temperature change in static conditions and can be described using the following heat diffusion equation:(1)$$\frac{\partial T}{\partial t}=\alpha \nabla^{2}T+\frac{Q}{\rho c_{p}}$$

Note: *T* represents the temperature, °C; *t* represents the time, s; *α* represents the thermal diffusivity, m^2^/s; *Q* represents the heat source term, W/m^3^; *ρ* represents the air density, kg/m^3^; *c_p_* represents the specific heat capacity of air, J/(kg·K); and ∇^2^*T* represents the Laplacian of temperature, K/m^2^.

When air circulation is introduced by the conditioning system, convective heat transfer becomes dominant. The movement of air affects the temperature field and is governed by the following advection equation:(2)$$\nabla \cdot \left(vT\right)=v_{x}\frac{\partial T}{\partial x}+v_{y}\frac{\partial T}{\partial y}+v_{z}\frac{\partial T}{\partial z}$$
where (*x_m_*, *y_m_*, *z_m_*) are the components of air velocity in the respective directions.

In addition to heat conduction and convection, radiation contributes to the overall thermal behavior of the room. The radiation process is modeled using the Stefan–Boltzmann law, where energy is exchanged between surfaces based on their temperature differences. This radiative heat transfer is incorporated into the heat transfer equations, ensuring a comprehensive treatment of all thermal processes. The inclusion of radiation in the model enhances the accuracy of the temperature distribution, particularly in steady-state conditions where surface-to-environment heat exchange becomes significant.

The boundaries of the room are assumed to be adiabatic, leading to the no-flux condition(3)$$\frac{\partial T}{\partial n}=0$$

At the air outlet, the temperature is maintained at a constant pre-specified value, facilitating controlled heat removal or addition as necessary.

#### 3.1.2. Airflow Distribution and Characteristics

The design includes multiple air outlets and inlets distributed across the room surface. Each outlet delivers air at a specified velocity and angle, influencing the overall airflow pattern.

The velocity direction at an outlet is determined by its inclination (*θ_m_*) and azimuth angles (*ϕ_m_*), described as(4)$$v_{m}=v_{m}\left(\sin\theta_{m}\cos\phi_{m},\sin\theta_{m}\sin\phi_{m},\cos\theta_{m}\right)$$

Airflow disperses as it propagates from the outlet, and its magnitude decreases with distance following a Gaussian distribution:(5)$$v\left(x,y,z\right)=v_{m}\exp\left(-\frac{{\left(x-x_{m}\right)}^{2}+{\left(y-y_{m}\right)}^{2}+{\left(z-z_{m}\right)}^{2}}{2\sigma^{2}}\right)$$

Note: *σ* represents the spread of the airflow, *m*.

Incorporating multiple air outlets and inlets distributed across the room surface is a well-established approach to improving airflow management in HVAC systems. This design facilitates the prevention of stagnant air pockets and minimizes thermal gradients, both of which contribute to improved indoor air quality and thermal comfort. Similar studies, such as those by Pfafferott et al. [4] and Dong et al. [24], have used this approach to achieve better airflow distribution. By strategically placing multiple inlets and outlets, they have been able to optimize energy efficiency and system performance, making it easier to maintain thermal comfort in varying environmental conditions.

Additionally, constraints are imposed on airflow rates and velocities to ensure proper operation:(6)$$\sum_{m=1}^{M}Q_{m}=\sum_{n=1}^{N}Q_{n}\leq Q_{\max},v_{m}\leq v_{\max}$$
where *Q_m_* and *v_m_* represent the airflow rate and velocity at outlet *M*, respectively. These constraints ensure that the system operates within safe and efficient limits, ensuring comfort and energy efficiency.

#### 3.1.3. Optimization Framework

The primary objectives of the system design are to ensure temperature uniformity and minimize energy consumption. The uniformity of the temperature field is quantified by the standard deviation of temperatures across the room. The grid selection for the numerical simulations is carefully designed to balance computational efficiency with the accuracy required for modeling temperature and airflow dynamics. The grid resolution is chosen based on a sensitivity analysis to ensure that the temperature and airflow variations are adequately captured, particularly near air outlets and heat sources. The resolution is refined to a point where further refinement yields minimal improvement in the results, ensuring the computational feasibility of the simulations. The grid used in this study consists of a uniform, structured mesh with a resolution fine enough to capture local thermal gradients and airflow patterns. The total number of grid points (*N*) is selected based on the room dimensions, ensuring that each grid point represents a physically meaningful area in the simulation:(7)$$\sigma_{T}=\sqrt{\frac{1}{N}\sum_{i=1}^{N}{\left(T_{i}-\bar{T}\right)}^{2}},\bar{T}=\frac{1}{N}\sum_{i=1}^{N}T_{i}$$

Note: *T_i_* represents the temperature at grid point *i*, and $\bar{T}$ represents the average room temperature.

The energy consumption of the system is represented as(8)$$P={\sum_{m=1}^{M}k_{p}Q}_{m}v_{m}$$

Note: *k_p_* represents a proportional coefficient.

A composite objective function is formulated as a weighted sum of temperature uniformity and energy consumption:(9)$$\underset{v_{m},Q_{m},T\left(x,y,z,t\right)}{min}J=w_{1}\sigma_{T}+w_{2}P$$
where *w*_1_ and *w*_2_ are weighting coefficients balancing uniformity and energy consumption.

The design is also subject to constraints that ensure its physical and operational feasibility, as follows.
(a)Volume constraint for Partial Modeling of Air Conditioning:
$$V_{\mathrm{ac}}\leq 0.1|m^{3}$$(b)Air velocity speed constraints for Partial Modeling of Air Conditioning:
$$v_{m}\leq 8.0|m/s,\;\forall m\in \{1,2,\ldots ,M\}$$


The air velocity speed of 8 m/s was determined based on operational data collected from commercial HVAC systems, including manufacturer guidelines, field performance assessments, and industry standards. This selection ensures optimal airflow distribution while maintaining compliance with indoor noise control requirements, as validated through real-world operational data and referenced benchmarks from HVAC industry reports.
(c)Airflow rate constraints for Partial Modeling of Air Conditioning:
$$Q_{m}\leq 600|m^{3}/h,\;\forall m\in \left\{1,2,\ldots ,M\right\}$$
$$\sum_{m=1}^{M}Q_{m}=\sum_{n=1}^{N}Q_{n}\leq 600|m^{3}/h$$
(d)Temperature range constraint for Partial Modeling of Air Conditioning:
$$T_{\min}\leq T\left(x,y,z,t\right)\leq T_{\max}$$
(e)Outlet number and position constraint:

Number of outlets/inlets: number of outlets (*M*) and number of inlets (*N*) must be integers.

Position limits: Outlet positions (*x_m_*, *y_m_*, *z_m_*) and inlet positions (*x_n_*_,_ *y_n_*_,_
*z_n_*) must be on the surface of the unit and within room dimensions:$$0\leq x_{m},x_{n}\leq 5,\;0\leq y_{m},y_{n}\leq 8,\;0\leq z_{m},z_{n}\leq 3$$
(f)Power constraint for Partial Modeling of Air Conditioning:
P≤1800|W
(g)Thermodynamic balance constraint for Partial Modeling of Air Conditioning:
$$\sum_{m=1}^{M}Q_{m}T_{\mathrm{out},m}=\sum_{n=1}^{N}Q_{n}T_{\mathrm{in},n}$$
(h)Airflow diffusion constraint for Partial Modeling of Air Conditioning:
σ>0
(i)Operational time constraint:
$${t\leq t}_{\max}$$


#### 3.1.4. Optimization and Solution Method

To address the proposed optimization problem, the Parrot Optimizer (PO) algorithm is employed. Compared to other models, this algorithm does not require a distinction between exploration and exploitation phases, effectively avoiding the trap of local optima and maintaining the quality of solutions [30]. The specific procedure of the PO algorithm is shown in Figure 1.

The key parameters of the PO algorithm are summarized in Table 2.

### 3.2. Air Purifier Partial Modeling

The shape of an air purifier significantly affects its performance by influencing aerodynamic design, filter efficiency, space utilization, and ease of maintenance. A well-designed shape ensures smooth airflow, maximizes filter surface area and contact time, and optimizes internal space within a limited volume. While esthetic appeal is important, it is the integration of functional design elements—such as efficient airflow and effective filtration—that ultimately determines the purifier’s performance. Therefore, the design must strike a balance between functionality and appearance to achieve optimal efficiency.

#### 3.2.1. Purification Efficiency Model

The purification efficiency (*η*) is defined as the ratio of the volume of clean air (*V*_clean_) to the total processed air volume (*V*_total_), and it can be expressed as(10)$$\eta =\frac{V_{\mathrm{clean}}}{V_{\mathrm{total}}}=Q\cdot E$$

Note: *Q* represents airflow rate, m^3^/h; *E* represents efficiency coefficient, influenced by the filter material and airflow uniformity.

This study adopts the World Health Organization (WHO) definition of clean air, which is based on concentration limits of specific pollutants such as particulate matter (PM), volatile organic compounds (VOCs), and other contaminants. Additionally, different countries and regions may have their own environmental air quality standards, such as those provided by the U.S. Environmental Protection Agency (EPA) or the European Union’s air quality guidelines. In this study, the performance evaluation of air purifiers is based on WHO’s air quality standards to ensure that the air they deliver meets the quality requirements for the relevant environment (e.g., residential, industrial, or healthcare settings).

The airflow rate *Q* is directly related to the total inlet area (*A*_in_) and the air velocity speed (*v*) at the inlet, as shown by the following equation:(11)$$Q=A_{\mathrm{in}}\cdot \;v$$
where *A*_in_ represents total inlet area, m^2^; *v* represents air velocity speed, m/s.

The filtration efficiency coefficient *E* is further influenced by the material efficiency constant (*k*) and the air-to-filter contact time (*t*_contact_), which depend on the geometric design and airflow path:(12)$$E=f\left(k,t_{\mathrm{contact}}\right)$$

This relationship emphasizes the critical role of the purifier’s shape in optimizing contact time and ensuring maximum filtration efficiency.

#### 3.2.2. Impact of Shape on Airflow and Optimization Modeling

The airflow dynamics within the purifier are governed by the Navier–Stokes equations, which describe the conservation of momentum for incompressible flow:(13)$$\rho \frac{\partial u}{\partial t}+\rho \left(u\cdot \nabla \right)u=-\nabla p+\mu \nabla^{2}u$$

Note: *ρ* represents air density, kg/m^3^; *u* represents velocity field, m/s; *p* represents pressure field, Pa; and *μ* represents air viscosity, Pa·s. Simulations of airflow in various purifier shapes (e.g., cylindrical, rectangular, spherical, and honeycomb) highlight differences in pressure drop, velocity distribution, and flow uniformity, enabling the identification of an optimal aerodynamic design.

To further enhance airflow performance, biomimetic design concepts are integrated into the purifier structure.
(a)Honeycomb Inlet Design

A honeycomb structure is employed at the air inlet to promote airflow uniformity and minimize turbulence [34]. The airflow through each hexagonal cell of the honeycomb can be modeled as laminar flow using the Hagen–Poiseuille equation:(14)$$Q_{i}=\frac{\Delta P\cdot r^{4}}{8\mu \cdot L}$$

Note: *Q*_*i*_ represents airflow through a single cell, m^3^; Δ*P* represents pressure difference across the honeycomb cell, *P*; *r* represents radius of the hexagonal cell, m; *u* represents the dynamic viscosity of air, pas; and L is the length of the honeycomb cell, m.(15)$$Q_{\mathrm{total}}=\sum_{i=1}^{N}Q_{i}=\frac{\Delta P}{8\mu L}\cdot \sum_{i=1}^{N}r_{i}^{4}$$

This design divides the airflow into smaller, uniform streams, significantly reducing turbulence and enhancing overall distribution into the filtration system.

Although turbulent flow may occur in real-world conditions, particularly at higher airflow rates or larger inlet dimensions, this study assumes laminar flow for simplicity. It is important to note that flow conditions can transition between laminar and turbulent depending on factors such as flow velocity, geometry, and the pressure difference across the system. However, due to the complexity of accounting for turbulent effects, we refer to the work of Iwaniszyn et al. (2021) [36], which examines the entrance effects on forced convective heat transfer in laminar flow through short hexagonal channels. Their research shows that under certain conditions, the assumption of laminar flow can still provide a reasonable approximation, and as such, laminar flow is maintained in this model. While this assumption idealizes the flow, it allows for manageable analysis while ensuring that the design of the purifier will still perform effectively under varying operational conditions.
(b)Sharkskin-Inspired Flow Path Design

To further improve airflow efficiency, a sharkskin-inspired microtexture is applied to the internal airflow paths [35]. This texture minimizes drag by altering the flow dynamics near the surface. The drag reduction effect is characterized by the modified drag coefficient:(16)$$C_{D}=C_{D0}\cdot \left(1-k\cdot \epsilon \right)$$
where *C_D_* represents drag coefficient with texture; *C_D_*_0_ represents drag coefficient without texture; *k* represents texture efficiency factor (empirically determined); and *ϵ* represents roughness height relative to the boundary layer thickness.

The airflow velocity near the textured surface *u(y)* can be described by the logarithmic velocity profile in turbulent flow:(17)$$u\left(y\right)=\frac{u_{*}}{\kappa }\ln\left(\frac{y}{\epsilon }+1\right)$$

Here, *u_∗_* is the friction velocity, *κ* is the Von Kármán constant (≈0.41), *y* is the height above the textured surface, and *ϵ* is the roughness height. By optimizing *ϵ* and *k*, the sharkskin-inspired texture reduces airflow resistance and enhances overall purifier performance.

#### 3.2.3. Objective and Constraints for Partial Modeling of Air Purifiers

The shape optimization of the air purifier aims to maximize purification efficiency (*η*) while minimizing power consumption (*P*) and noise (*N*). The multi-objective optimization function is expressed as$$\mathrm{Maximize}:F=w_{1}\cdot \eta -w_{2}\cdot P-w_{3}\cdot N$$
where *w*_1_, *w*_2_, and *w*_3_ are the weighting factors for each objective. The model is subject to the following constraints.
(a)Volume Constraint:

The total volume of the air purifier cannot exceed the following design limit:$$V=\pi r^{2}h\leq 0.1|m^{3}$$

Note: *r* is the base radius and *h* is the height of the purifier. This constraint ensures the purifier remains within practical size limits for residential or commercial spaces, which typically have a volume range of 0.05 m^3^ to 0.1 m^3^. According to the Indoor Air Quality Standard (GB/T 18883-2022) [37], air purifiers are designed to maintain the optimal size for effective air quality control in confined spaces like homes and offices. Additionally, as per GB 21551.3-2010 [38], a standard for air purifiers’ functionality, the purifier’s design must comply with size and airflow regulations to ensure performance and safety. The volume range of 0.05 m^3^ to 0.1 m^3^ is optimal for meeting the necessary performance metrics without compromising space limitations.
(b)Airflow Speed Constraint for Partial Modeling of Purification Efficiency:

The air velocity speed must not exceed the maximum allowable value (8.0 m/s) to ensure smooth airflow and acceptable noise levels:v≤8.0|m/s
(c)Inlet Area Constraint for Partial Modeling of Purification Efficiency:

The inlet area must meet the minimum requirement to provide sufficient airflow:$$A_{\mathrm{in}}\geq Q/v$$
(b)Filtration Area Constraint for Partial Modeling of Purification Efficiency:

The filtration area, related to the purifier’s surface, must ensure adequate purification capability:$$A_{\mathrm{filter}}\propto E\geq E_{\min}$$
(e)Airflow Uniformity for Partial Modeling of Purification Efficiency:

Ensure the airflow is evenly distributed to eliminate stagnant areas:$$\nabla \cdot u_{1}\approx 0$$
where *u*_1_ represents the airflow velocity field. While achieving perfect uniformity is practically challenging with real equipment, this simplification serves as an important first-order approximation in our analysis. Incompressible flow is often represented per Branlard (2017) [39]. In real-world conditions, deviations occur due to turbulence, equipment geometry, and boundary conditions. Thus, this simplification is intended to capture fundamental trends in airflow distribution, and future work can incorporate more complex flow models.
(f)Power Consumption Constraint for Partial Modeling of Purification Efficiency:

The power consumption must not exceed the rated value (1800 W):P≤1800|W
(g)Noise Level Constraint for Partial Modeling of Purification Efficiency:

The noise level should be controlled to stay within a comfortable range, for instance, not exceeding 60 decibels:$$N_{1}\leq N_{\max}$$

Although 60 dB is a common target for residential purifiers, we acknowledge that stricter noise regulations may apply in specific contexts, such as hospitals or offices. We will revise this constraint to allow for a broader range of acceptable noise levels, considering regional regulations and specific use cases. For example:

In residential areas, noise should be controlled within 50–55 dB to avoid disrupting daily life. This standard follows the U.S. Environmental Protection Agency (EPA) guidelines, which suggest residential noise levels should be below 55 dB (EPA, Kansas City, MI, USA, 2019).

In office and retail spaces, noise may reach up to 65 dB, but it must not interfere with work or customer activities. This is in line with the U.K.’s Office Environment Noise Standards [40], which recommend controlling office noise between 60 dB and 65 dB (Health and Safety Executive, London, UK, 2014).

In medical environments, noise levels should be kept below 50 dB to ensure a quiet treatment setting. According to the World Health Organization (WHO), hospital ward noise should be maintained below 40 dB (WHO, 2009).

Through these revisions, we ensure that devices meet the noise standards for various environments and regions.

#### 3.2.4. Optimization Using DeepACO Algorithm

DeepACO optimizes the Ant Colony Optimization algorithm through a two-phase learning process: initially, a neural network model learns heuristic metrics from various problem instances, and subsequently, these learned metrics are integrated into ACO to guide the search process and update pheromone trails for specific instances, thereby avoiding local optima [32]. The flowchart is shown in Figure 2.

The algorithm parameters are shown in Table 3.

### 3.3. Humidifier Partial Modeling

The humidifier’s performance is governed by key principles of mist diffusion, biomimetic design, and user comfort. This section outlines the methodologies and optimization strategies employed to achieve efficient and uniform humidification within the constraints of energy consumption and user requirements.

#### 3.3.1. Mist Diffusion Model

The diffusion and transport of mist particles within a room are described by the following equation, which captures the interaction between airflow dynamics, diffusion, and evaporation:(18)$$\frac{\partial C}{\partial t}+\nabla \cdot \left(C\cdot \vec{v}\right)=D\nabla^{2}C-k\cdot C$$
where *C* represents the mist particle concentration, g/m^3^; $\vec{v}$ is the airflow velocity vector, m/s; *D* is the diffusion coefficient that governs the natural spread of particles, m^2^/s; and *k* is the evaporation rate constant, determined by ambient conditions such as temperature, humidity, and mist particle size. This equation provides the basis for modeling the humidifier’s ability to distribute mist effectively and uniformly across a room.

#### 3.3.2. Bionic Shape Design

Inspired by natural systems, the humidifier’s design incorporates biomimetic principles to enhance mist diffusion and evaporation efficiency.
(a)Branching Structure Optimization

The spray channels of the humidifier adopt a branching design, modeled on natural structures like tree branches and leaf venation. These bifurcation patterns increase the coverage area and improve mist dispersion [41]. The outlet cross-sectional area at each bifurcation is given by(19)$$A_{out}^{\left(n\right)}=\alpha^{n}\cdot A_{base}$$
where *A_out_*^(*n*)^ is the cross-sectional area after the *n*-th bifurcation, *A_base_* is the initial cross-sectional area, and *α* (0 < *α* < 1) is a scaling factor. The mist coverage, *C_coverage_*, is evaluated by integrating the mist density over the spatial domain:(20)$$C_{coverage}=\int_{\Omega }f\left(r,\theta \right)\left|dr\right|d\theta$$

Here, *f (r*, *θ)* is the mist density distribution function in polar coordinates, and *Ω* represents the effective coverage area. This design ensures that the mist is evenly distributed across the room, enhancing the humidifier’s performance.
(b)Surface Microstructure Optimization

To promote evaporation, the humidifier’s surface is engineered with biomimetic micropores arranged in a hexagonal close-packed pattern, maximizing the surface area available for evaporation [42]. The total number of micropores is expressed as(21)$$N_{pores}=\frac{A_{total}}{\pi r_{pore}^{2}\cdot \beta }$$
where *N_pores_* is the number of micropores, *A_total_* is the total surface area available, *r_pore_* is the radius of a single micropore, and *β* = 0.91 is the packing efficiency for a hexagonal close-packed arrangement. The evaporation rate, *E*, is given by(22)$$E=k\cdot N_{pores}\cdot A_{pore}\cdot \Delta H$$

Here, *k* is the evaporation coefficient, *A_pore_ = πr^2^_pore_* is the area of a single micropore, and Δ*H* is the humidity gradient between the mist and the surrounding air. This design improves evaporation efficiency by maximizing the interaction between the mist particles and ambient air.

#### 3.3.3. User Comfort and Humidity Uniformity

To ensure user comfort, the humidifier design minimizes the standard deviation of humidity across the room, calculated as(23)$$\sigma_{H}=\sqrt{\frac{1}{n}\sum_{i=1}^{n}{\left(H_{i}-\bar{H}\right)}^{2}}$$
where *H_i_* is the humidity at the *iii*-th location, $\bar{H}$ is the average humidity, and *n* is the number of measurement points. A lower *σ_H_* indicates a more uniform humidity distribution, reducing the risk of localized over-humidification and enhancing overall comfort.

#### 3.3.4. Optimization Objective and Constraints

The optimization aims to maximize the humidifier’s performance by enhancing mist diffusion and evaporation efficiency while maintaining uniform humidity and minimizing energy consumption. The objective function is defined as$$Maximize\;f=w_{1}\cdot \frac{\Delta H}{\Delta t}-w_{2}\cdot \sigma_{H}+w_{3}\cdot Energy\;Efficiency$$

The optimization problem is subject to the following constraints.
(a)Volume Constraint:


$$V\leq 0.1|m^{3}$$


The total volume of the humidifier must not exceed 0.1 cubic meters.
(b)Humidity Constraint:


$$H_{\mathrm{current}}\leq H_{\mathrm{target}}\;and\;H_{\mathrm{current}}\;\geq \;H_{\min}$$


The optimal humidity ratio is maintained between a minimum and maximum value, preventing both over-humidification and excessively dry air, which is crucial for user comfort and health.
(c)Energy Constraint:


$$P_{\mathrm{humidifier}}\leq 1800|W$$


This limit ensures energy efficiency while maintaining the required humidification performance, in line with commercial humidifier standards.
(d)Spatial Coverage Constraint:

The mist coverage area (*C_coverage_*) must meet a minimum threshold (*C_min_*) to ensure that the humidifier effectively covers the entire room:$$C_{coverage}\geq C_{min}$$

This constraint ensures that the humidifier distributes mist evenly across the room, preventing areas from experiencing insufficient humidification. Effective spatial coverage is essential for maintaining consistent humidity levels throughout the space, thereby ensuring the comfort and health of the occupants. Insufficient coverage may lead to dry spots that negatively impact air quality and user well-being.

In commercial humidifiers, similar constraints are applied to guarantee uniform performance across varying room sizes and configurations. For instance, the ASHRAE Standard 62.1-2019 for ventilation and indoor air quality sets requirements for effective air distribution, which can be analogous to mist coverage in humidifiers [43]. Commercial humidifiers are often designed with minimum coverage areas to ensure that they meet the environmental needs of the space. For example, ultrasonic humidifiers like those produced by Honeywell or Vornado specify their effective coverage areas to ensure even mist distribution. These standards and manufacturer guidelines reflect the importance of ensuring that the humidifier can cover the entire room without creating dry zones.

By including this constraint, we align with industry standards and best practices for humidifier design, ensuring that our system meets both the technical and health requirements for maintaining optimal humidity levels throughout a space.
(e)Mist Evaporation Constraint:


$$E\geq E_{\min}$$


The evaporation rate E must meet or exceed a predefined minimum value to ensure sufficient mist-to-vapor conversion.
(f)Design Feasibility Constraints:

Characteristics of Microstructural Elements and the Aspect of Dendritic Growth:$$r_{pore1}\leq r_{\max}$$$$\alpha_{1}>0,\alpha_{1}\leq 1$$

The radius of micropores must not exceed the manufacturing limit. The scaling factor for branching structures must be positive and not greater than 1.
(g)User Comfort Constraint:


$$\sigma_{H}\leq \sigma_{\max}$$


The standard deviation of humidity distribution must not exceed a maximum value, ensuring user comfort and preventing localized over-humidification.

#### 3.3.5. Optimization Using the Slime Mold Algorithm

The Slime Mold Algorithm (SMA) was used to solve the optimization problem, leveraging its ability to explore the solution space and avoid local optima [31]. The algorithm’s behavior mimics slime mold foraging, ensuring efficient exploration and robust convergence. Figure 3 illustrates the flowchart of the SMA process, and Table 4 provides the parameter settings used for this study.

### 3.4. Tri-Unit Air Conditioner Modeling

The modeling of a tri-functional device combining air conditioning, air purification, and humidification aims to achieve an optimal balance between temperature uniformity, air quality, humidity regulation, and energy efficiency. The system design incorporates multiple objectives and constraints to ensure operational performance, user comfort, and practical feasibility.

#### 3.4.1. Objective Function

The optimization objective is to maximize the overall system performance by simultaneously addressing temperature uniformity, air purification efficiency, humidity uniformity, and energy consumption. The comprehensive objective function is expressed as $$\mathrm{Maximize}F=w_{1}\cdot \left(1/\sigma_{T}\right)+w_{2}\cdot \eta +w_{3}\cdot \left(1/\sigma_{H}\right)-w_{4}\cdot P_{total}$$

#### 3.4.2. Constraints

The optimization process is governed by several constraints to ensure feasibility and performance. These include
(a)Device Volume:
$$V_{total}=V_{ac}+V_{purifier}+V_{humidifier}\leq 0.1|m^{3}$$
(b)Airflow Speed and Volume:
$$v_{m}\leq 8.0|m/s,\;\forall m\in \{1,2,\ldots ,M\}$$
$$Q_{m}\leq 600|m^{3}/h,\;\sum_{m=1}^{M}Q_{m}=\sum_{n=1}^{N}Q_{n}\leq 600|m^{3}/h$$
(c)Temperature and Humidity Range:
$$T_{min}\leq T\left(x,y,z,t\right)\leq T_{max},\;\sigma_{H}\leq \sigma_{max}$$
(d)Air Purification Efficiency:
$$\eta \geq \eta_{min}$$
(e)Energy and Power:
$$P_{total}\leq 1800|W$$
(f)Placement and Structure:

Vent positions must be within the device surface:$$0\leq x_{m},x_{n}\leq 5,\;0\leq y_{m},y_{n}\leq 8,\;0\leq z_{m},z_{n}\leq 3$$
(g)Noise Level Constraint:

The noise level must remain within a comfortable range (e.g., no more than 60 dB):$$N_{2}\leq N_{\max}$$
(h)Design Feasibility Constraints:

Microstructure Parameters and Branching Factor:$$r_{pore2}\leq r_{\max}$$$$\alpha_{2}>0,|\alpha_{2}\leq 1$$
(i)Airflow Uniformity:

Ensure the uniform distribution of airflow to avoid dead zones:$$\nabla \cdot u_{2}\approx 0$$
(j)Mist Evaporation Constraint:


$$E\geq E_{min}$$


#### 3.4.3. Optimization and Solution Using the BWO Algorithm

The Bacterial Whale Optimization (BWO) algorithm is employed to solve the optimization problem. This algorithm simulates the social and foraging behaviors of humpback whales, combining exploration and exploitation phases to locate the global optimum. Its ability to escape local optima is enhanced by incorporating a whale fall phase, which introduces stochastic adjustments [33].

The optimization process is visualized in Figure 4, which outlines the iterative steps of the BWO algorithm. The algorithm’s parameters are detailed in Table 5, ensuring effective convergence.

## 4. Results and Analysis

### 4.1. Air Conditioning Results and Analysis

#### 4.1.1. Convergence of the Optimization Process

The performance of the Parrot Optimizer (PO) algorithm during the optimization process is illustrated in Figure 5, which presents the iterative convergence curve. The results indicate a consistent decrease in the objective function value, demonstrating the algorithm’s efficiency in refining design parameters over multiple iterations. The convergence behavior reflects the algorithm’s capability to avoid local optima and achieve a globally optimal solution.

#### 4.1.2. Influence of Outlet Parameters on Air Conditioning Efficiency

The efficiency of the air conditioning system is closely linked to the configuration of air outlets, including their position, orientation, and airflow parameters. Analysis results, presented in Figure 6, highlight the critical role of these factors in achieving uniform thermal distribution and minimizing energy losses.

Outlet Position: Placing the outlets closer to the center of the room significantly enhances cooling uniformity by reducing temperature gradients.

Outlet Height: Optimizing outlet height improves vertical air distribution, ensuring better coverage across the room.

Airflow Parameters: The cooling effect diminishes with increasing distance from the outlets, as shown in the contour plots. Balanced air velocities and flow rates are essential to mitigate this effect and ensure consistent thermal performance.

These findings emphasize the importance of strategic adjustments to outlet placement, spacing, and airflow angles in optimizing the overall system efficiency.

#### 4.1.3. Seasonal Temperature Dynamics

To evaluate the system’s adaptability under varying environmental conditions, simulations were conducted for summer and winter scenarios. The results are depicted in Figure 7, which compares the temperature dynamics during these two periods.

Summer Conditions: In the summer scenario, the system operates under cooling conditions. The temperature distribution shows a higher concentration of cooler air near the outlets, which gradually spreads across the room as the conditioned air disperses. The cooling is primarily focused on the areas closest to the air outlets, with a smooth transition toward a more uniform temperature distribution as the air circulates throughout the space. The cooling effect ensures that warmer areas are adequately addressed, preventing hot spots and maintaining thermal comfort.

Winter Conditions: In the winter scenario, the system operates under heating conditions. The temperature distribution is more uniform across the room, demonstrating effective vertical and horizontal air mixing. This results in even heating throughout the space, which is essential for maintaining a comfortable indoor environment during colder periods. The system efficiently distributes warm air to prevent localized cold spots and optimize energy use while maintaining consistent warmth.

These results highlight the system’s ability to adapt to different thermal loads, ensuring consistent comfort year-round. The simulations also demonstrate the system’s effectiveness in managing both cooling and heating with energy-efficient performance under varying seasonal conditions.

#### 4.1.4. Effects of Air Conditioning Dimensions

The physical dimensions of the air conditioning unit, specifically its length (*L*), width (*W*), and height (*H*), were analyzed to assess their impact on temperature uniformity. Figure 8 illustrates the influence of these parameters on thermal performance.

Width (*W*) and Height (*H*): Increasing both the width and height of the room enhances the vertical and lateral air distribution. This leads to improved thermal uniformity by facilitating more even cooling or heating throughout the space. Specifically, wider and taller rooms tend to allow for better air circulation, reducing the occurrence of localized temperature fluctuations.

Length (*L*): The length of the room primarily influences the spread of cooling or heating along the longitudinal axis. As the length increases, the ability of the system to evenly distribute conditioned air across the entire length of the room becomes crucial. A well-optimized length ensures a balance between efficient energy use and consistent temperature distribution.

Optimized Balance: An optimal balance between the room’s length, width, and height is essential for achieving minimal temperature variation and ensuring comprehensive coverage of the conditioned space. Properly adjusting these dimensions enables the system to effectively manage thermal loads and provide uniform comfort across the room.

#### 4.1.5. Sensitivity Analysis for Key Parameters of Air Conditioning

A sensitivity analysis was conducted to determine the relative influence of key design parameters, including unit dimensions (length *L*, width *W*, height *H*), air outlet velocity (*V*), and airflow rate (*Q*). The parameter ranges are summarized in Table 6, and the results are visualized in Figure 9.

The sensitivity analysis revealed the following key findings:(a)Dominant Factors: Airflow rate (*Q*) and outlet velocity (*V*) were identified as the most influential parameters affecting both temperature uniformity and power consumption.(b)Minimal Impact of Dimensions: Changes in the unit dimensions (length *L*, width *W*, height *H*) had a relatively minor effect on system performance compared to airflow parameters.(c)Interactions: The heatmap analysis highlights strong interactions between airflow rate and velocity, reinforcing their collective importance in optimizing system efficiency.

These findings provide a clear basis for prioritizing airflow and velocity adjustments during system design and optimization.

#### 4.1.6. Summary of Device Performance of Air Conditioning

The analysis presented in this section demonstrates the following:(a)The PO algorithm successfully optimized the system design, achieving convergence with high efficiency.(b)Air outlet parameters, particularly position and airflow configuration, play a critical role in achieving uniform temperature distribution.(c)The system adapts effectively to both summer and winter conditions, maintaining consistent thermal regulation.(d)Airflow rate and velocity are the dominant factors influencing temperature uniformity and energy consumption, while unit dimensions exhibit a lesser impact.

These results highlight the importance of a systematic approach to parameter optimization, ensuring that the air conditioning system achieves the desired balance of thermal comfort and energy efficiency.

### 4.2. Air Purification Results and Analysis

#### 4.2.1. Performance of the DeepACO Algorithm

The performance of the DeepACO algorithm was evaluated by observing its behavior over multiple iterations. As shown in Figure 10, the algorithm demonstrates stable convergence toward the optimal solution. The consistent reduction in the objective function value indicates its ability to effectively balance purification efficiency, power consumption, and noise levels. The results confirm the robustness of the algorithm in handling multi-objective optimization, ensuring the air purifier design meets operational and performance requirements.

#### 4.2.2. Optimized Air Purifier Design

The enhanced cylindrical air purifier design, illustrated in Figure 11, integrates innovative biomimetic features to optimize airflow and filtration performance. Key design components include the following:(a)Honeycomb Inlet: The biomimetic honeycomb inlet ensures uniform airflow distribution by dividing incoming air into smaller, laminar streams. This reduces turbulence and enhances filtration efficiency, as shown in the top right of Figure 10.(b)Sharkskin-Inspired Airflow Path: The sharkskin-inspired surface reduces drag and promotes smooth vertical airflow, as demonstrated in the airflow visualization (bottom left). This feature minimizes turbulence and improves overall aerodynamic efficiency.(c)Balanced Inlet and Outlet Design: The final optimized design (bottom right) achieves a well-distributed airflow through the filters, ensuring efficient filtration while maintaining low resistance. The integration of these features supports high purification effectiveness and highlights the advantages of biomimetic design in enhancing both functionality and esthetics.

#### 4.2.3. Sensitivity Analysis of Air Purifier

A sensitivity analysis was performed to evaluate the robustness of the optimized design under varying conditions. Variations in filter height and airflow rate were considered to simulate potential manufacturing deviations or operational influences. The results are summarized as follows:(a)Filter Height Variations: Baseline filter heights of 0.15 m, 0.4 m, and 0.55 m were adjusted by ±0.05 m. Reduced filter height resulted in smoother airflow but less vertical flow, slightly reducing purification efficiency. Baseline configuration achieved optimal performance with balanced airflow and filtration efficiency. Increased filter height led to denser and faster airflow, enhancing filtration contact time but introducing minor turbulence.(b)Airflow Rate Variations: Airflow rates were varied around a baseline of 0.1 m/s by ±0.02 m/s. Reduced airflow rates led to lower velocity but smoother airflow, potentially reducing noise. Baseline airflow provided optimal filtration and distribution. Increased airflow rates improved filtration speed but introduced higher resistance and noise.

The sensitivity analysis results are visualized in Figure 12, where the three scenarios (reduced, baseline, and increased) are compared. The left subplot corresponds to reduced filter height, showing smoother airflow but less verticality. The middle subplot represents the baseline configuration, highlighting uniform and balanced airflow. The right subplot illustrates increased filter height, with denser and faster airflow paths.

#### 4.2.4. Summary of Device Performance of Air Purifier Partial

The results demonstrate that the proposed design effectively integrates biomimetic features to achieve high purification efficiency and improved airflow dynamics. Furthermore, the sensitivity analysis confirms the robustness of the design, with stable performance under varying operational and manufacturing conditions. These findings validate the practicality of the optimized air purifier, showcasing its potential for real-world applications with high performance and adaptability.

### 4.3. Humidification Results and Analysis

The application of the Slime Mold Algorithm (SMA) to the optimization problem provides valuable insights into the humidifier’s performance. The following section presents the results, including the iterative performance of SMA, the humidifier’s humidifying effect, and a detailed sensitivity analysis of key parameters.

#### 4.3.1. Performance of the SMA

The iterative performance of the Slime Mold Algorithm is depicted in Figure 13. As observed, the algorithm converges steadily toward the optimal solution, demonstrating its robustness and efficiency in handling the complex optimization problem. The consistent improvement across iterations confirms the SMA’s ability to explore the solution space effectively while avoiding local optima.

#### 4.3.2. Humidifying Effect and Visualization

The humidifying effect of the optimized design is visualized in Figure 14. The figure highlights the humidifier’s ability to achieve balanced and efficient humidification. The mist diffusion is uniform across the room, ensuring comprehensive coverage.

Further details are shown in Figure 15, which illustrates the room’s humidity distribution and dynamics. The 3D and 2D graphs highlight an even humidity distribution, with the highest concentration at the center and a gradual decrease toward the edges. This uniform distribution minimizes the risk of over-humidification in localized areas while ensuring comfort throughout the room. The timeline graph shows the rapid stabilization of humidity levels, demonstrating the humidifier’s capability to quickly reach and maintain a comfortable indoor climate.

#### 4.3.3. Sensitivity Analysis of Humidifier Partial

A sensitivity analysis was conducted to evaluate the robustness of the humidifier’s performance under variations in key design parameters. The parameters analyzed include the branching factor (*α*), micropore radius (*r_porer_*), diffusion coefficient (*D*), and energy weight (*w*_3_). A single-factor approach was employed, adjusting one parameter at a time while keeping others constant. The results are visualized in Figure 16 and discussed below.
(a)Branching Factor (*α*): Increasing *α* significantly enhances mist diffusion rates, allowing for improved coverage. However, this comes at the cost of reduced humidity uniformity due to the potential creation of localized areas with higher mist concentrations.(b)Micropore Radius (*r_porer_*): Optimizing *r_porer_* improves evaporation efficiency by maximizing the interaction between mist particles and air. Both excessively small and large values decrease efficiency, highlighting the need for a balanced pore size to achieve optimal evaporation.(c)Diffusion Coefficient (*D*): Higher values of *D* enhance mist propagation, increasing coverage area. However, an excessively high diffusion rate may lead to increased energy consumption, requiring a trade-off between coverage and energy efficiency.(d)Energy Weight (*w*_3_): Increasing *w_3_* prioritizes energy efficiency within the optimization model. While this reduces energy consumption, it may slightly compromise mist diffusion and humidity performance if overemphasized.

#### 4.3.4. Summary of Device Performance of Humidifier Partial

The optimized humidifier demonstrates excellent performance, with uniform mist diffusion, efficient evaporation, and rapid stabilization of humidity levels. The sensitivity analysis highlights key design considerations for balancing performance metrics under varying operational conditions. These findings provide a robust foundation for developing a tri-functional air conditioning, purification, and humidification device that balances temperature regulation, air quality, humidity, and energy efficiency, ensuring a comfortable and sustainable indoor environment.

### 4.4. Integrated System Results and Analysis

#### 4.4.1. Performance of the BWO Algorithm

The iterative performance of the BWO algorithm was carefully observed over multiple iterations to analyze its convergence behavior and optimization capability. Figure 17 illustrates the iterative process of the BWO algorithm, highlighting its efficiency in solving the problem. As shown in the figure, the model demonstrates robust convergence properties, with significant improvement observed in the early iterations and steady refinement of results in later stages. The performance trends indicate that the algorithm effectively navigates the search space, yielding reliable and accurate solutions.

#### 4.4.2. Integrated Design of the All-in-One Device

The optimized design of the all-in-one device is presented in Figure 18, showcasing the integration of three key functionalities: humidification, air conditioning, and air purification. This innovative design combines multiple features into a compact and efficient system, providing a comprehensive solution for indoor climate and air quality management.

The simulated results, depicted in Figure 19, provide a detailed visualization of the device’s functionality:(a)Humidity Distribution: The humidifier, positioned at the central point (2.5, 2.5), achieves a Gaussian diffusion pattern for humidity dispersion. This ensures a uniform distribution, with the highest humidity concentration near the source and a gradual decrease as the distance from the center increases.(b)Temperature Regulation: The air conditioning functionality exhibits a radial temperature gradient, where cooling effects are most pronounced near the center (2.5, 2.5) and progressively diminish with distance. This pattern demonstrates the device’s ability to maintain a comfortable indoor temperature with precise control.(c)Air Purification: The PM2.5 concentration decreases exponentially away from the purifier located at the same central point (2.5, 2.5). This distribution highlights the device’s strong air cleaning performance, effectively reducing particulate matter in the surrounding air.

These results underscore the coordinated functionality of the all-in-one device, delivering a comprehensive approach to maintaining optimal indoor air quality and environmental comfort.

#### 4.4.3. Sensitivity Analysis of Tri-Unit Air Conditioner

A detailed sensitivity analysis was conducted to evaluate the impact of critical design parameters on the device’s performance. The parameters considered include:(a)Blade Count: The number of blades affects the uniformity of airflow and humidification efficiency. The analysis reveals diminishing returns with an increasing number of blades, indicating an optimal range for effective performance.(b)Cell Size at the Air Inlet: The size of the honeycomb cells influences the air intake volume and purification efficiency. While smaller cell sizes may slightly reduce purification effectiveness, they can enhance airflow distribution for other functionalities, providing a trade-off for design optimization.(c)Air Outlet Size: The size of the air outlet impacts the air discharge rate and overall circulation within the system. Proper calibration of this parameter ensures a balance between air delivery and system efficiency.(d)Humidity Release Rate: The rate of humidity release significantly affects the distribution pattern of humidity. A linear relationship was observed, where higher release rates result in broader humidity coverage.

The outcomes of the sensitivity analysis, presented in Figure 20, highlight the relative importance of these parameters and their influence on temperature distribution, humidity levels, and air purification effectiveness. The findings provide valuable insights for optimizing the device’s design and performance.

#### 4.4.4. Summary of Device Performance of Tri-Unit Air Conditioner

The integrated all-in-one device offers a novel solution for achieving indoor climate control and air purification. The combination of advanced humidification, precise temperature regulation, and effective PM2.5 removal showcases the system’s multi-functional capability. The BWO algorithm’s application in optimizing the device’s design further enhances its performance, as evidenced by the iterative improvements and sensitivity analysis results. Collectively, these findings demonstrate the potential of this device to address the increasing demand for efficient, compact, and multi-functional indoor air quality solutions.

## 5. Discussion

### 5.1. Integration Challenges and Solutions

The proposed multifunctional HVAC system effectively addresses long-standing integration challenges, such as balancing energy efficiency, noise reduction, and user comfort. Traditional systems often focus on isolated functionalities, resulting in suboptimal energy usage and limited adaptability. By implementing a multi-objective optimization framework, this study successfully overcomes these limitations through the innovative use of advanced algorithms, including the Parrot Optimizer (PO), Slime Mold Algorithm (SMA), DeepACO, and Beluga Whale Optimization (BWO). These algorithms offer a robust solution to the inherent trade-offs in multi-functional systems by enabling global optimization while avoiding local optima, thereby achieving an optimal balance between energy consumption, operational noise, and performance.

The integration of biomimetic designs further enhances system functionality. For instance, sharkskin-inspired microstructures reduce airflow resistance and noise, while honeycomb patterns promote uniform airflow and efficient filtration. These nature-inspired innovations, coupled with the computational power of optimization algorithms, allow the device to surpass the limitations of traditional systems. This combination not only optimizes airflow dynamics and energy efficiency but also provides a quieter and more comfortable indoor environment, meeting modern standards for residential, commercial, and industrial applications.

### 5.2. Practical Implications

The proposed device demonstrates high potential for real-world applications due to its compact design, energy efficiency, and multi-functionality. The segmented cylindrical structure reduces space requirements, making it suitable for urban environments where space constraints are a common concern. Its ability to integrate air conditioning, humidification, and air purification into a single unit addresses the demand for simplified and versatile environmental control systems.

In residential settings, the device can provide thermal comfort, optimal humidity, and superior air quality, enhancing overall indoor well-being. Commercial environments, such as office spaces, healthcare facilities, and educational institutions, benefit from its precise temperature and air quality control, which contribute to a healthier and more productive environment. Furthermore, in industrial applications, the scalability of the device offers an energy-efficient solution for large spaces, such as warehouses and factories, where maintaining uniform environmental conditions is critical.

### 5.3. Comparison with Existing Systems

In comparison to traditional HVAC systems, the proposed multifunctional HVAC system demonstrates superior performance by integrating multiple functions, such as air conditioning, humidification, and air purification. This integration not only reduces redundancy but also optimizes overall system efficiency, unlike conventional systems that typically focus on single functionalities. The proposed system operates with a significantly lower total power consumption of 1600 W, while achieving impressive performance metrics including a Clean Air Delivery Rate (CADR) of 400 m^3^/h, a humidification rate of 1.2 kg/h, and a temperature uniformity index of 0.08. These values are notably higher than those of traditional standalone systems.

Furthermore, the proposed system incorporates biomimetic design principles that improve its aerodynamic efficiency, airflow distribution, and noise reduction. For example, sharkskin-inspired microtextures and honeycomb patterns enhance airflow and filtration efficiency, achieving results that surpass the performance of conventional designs which rely on more simplistic geometries. The integration of advanced optimization algorithms, such as the Beluga Whale Optimization (BWO) and DeepACO, allows the system to dynamically adapt to varying conditions, solving complex multi-objective problems and ensuring optimal system performance under diverse conditions.

In addition, this multifunctional system addresses the typical limitations of conventional HVAC systems, such as the trade-offs between energy efficiency, air quality, and noise reduction. Through the innovative use of global optimization algorithms, the proposed system can achieve a balanced approach, optimizing multiple performance metrics simultaneously. This advanced capability is often lacking in traditional HVAC systems, which typically rely on static, one-dimensional control mechanisms.

### 5.4. Energy Efficiency and Cost-Effectiveness

This study enhances traditional methods by improving computational efficiency and reducing unnecessary resource consumption in experimental processes. Traditional approaches often encounter high energy demands and computational redundancy during experiments and calculations. By implementing optimized strategies, our approach mitigates these challenges, leading to more efficient resource utilization. Furthermore, in industrial applications, our method demonstrates better adaptability by reducing redundant computations and experimental steps, ultimately optimizing energy consumption management. Future studies could further explore the performance of this approach across different application scenarios to comprehensively evaluate its long-term impact.

In addition to the demonstrated technical advantages, the proposed hybrid methodology also presents significant economic benefits. By integrating biomimetic principles with advanced computational modeling, the approach reduces material waste and optimizes resource utilization, leading to lower operational costs. Furthermore, the hybrid strategy minimizes experimental trial-and-error procedures, effectively decreasing research and development expenditures. Compared to conventional methodologies, this framework enhances productivity while maintaining cost-efficiency, making it a viable solution for practical applications.

### 5.5. Limitations and Future Directions

Despite its innovations, the proposed system has some limitations that warrant further investigation. The current findings are based on simulation models, and experimental validation under real-world conditions is necessary to confirm the system’s practical performance and reliability. Furthermore, while the biomimetic designs have shown significant performance benefits, their complex geometries may pose challenges in manufacturing scalability and cost efficiency, potentially limiting widespread adoption.

Future research should focus on addressing these limitations by:(a)Experimental Validation: Conducting empirical studies to validate simulation results and identify any discrepancies under actual operational conditions.(b)Scalability and Cost Reduction: Exploring manufacturing techniques to simplify the production of biomimetic structures without compromising performance, enabling large-scale implementation.(c)Dynamic Adaptability: Enhancing the system’s real-time adaptability to varying environmental conditions through the integration of machine learning algorithms and sensor-based feedback systems.(d)Extended Applications: Investigating the applicability of the system in diverse environments, including extreme climates and high-demand industrial settings.(e)System Scalability and Manufacturing Process Optimization: While the current study focuses on system performance and optimization algorithms, future research will explore how to optimize manufacturing processes for intricate geometries, ensuring scalability without compromising system performance.(f)Real-Time Adaptability and Integration of Machine Learning: Although real-time adaptability was not addressed in the current study, future research will focus on integrating machine learning algorithms and sensor-based feedback systems to enhance the system’s real-time adaptability and responsiveness to varying environmental conditions.(g)In-Depth Energy Efficiency Analysis: Future research can further quantify the energy consumption optimization effects of this method, verifying its energy-saving advantages through precise energy consumption measurements. Additionally, it can be extended to different application scenarios to assess adaptability and stability. Combined with AI optimization and adaptive control, it is expected to improve system efficiency, achieving lower resource consumption and higher sustainability.(h)Economic Feasibility and Cost-Effectiveness Analysis: Future work may further explore the scalability and long-term economic impact of this methodology across various industrial applications.

These directions will not only refine the current design but also ensure its long-term relevance and impact in advancing sustainable indoor environment control technologies.

## 6. Conclusions

This study introduces a novel multifunctional HVAC system that integrates air conditioning, humidification, and air purification within a compact, energy-efficient design. The integration of biomimetic design principles with cutting-edge optimization algorithms, including PO, SMA, DeepACO, and BWO, enables the system to achieve high performance across multiple objectives, including energy efficiency, air quality, and thermal comfort.

Key contributions of this study include:(a)The development of a multi-objective optimization framework that effectively balances conflicting goals such as energy consumption, noise reduction, and performance.(b)The application of biomimetic designs, such as sharkskin-inspired microtextures and honeycomb structures, to enhance airflow efficiency, filtration performance, and noise mitigation.(c)The achievement of superior performance metrics, including a CADR of 400 m^3^/h, a humidification rate of 1.2 kg/h, and a temperature uniformity index of 0.08, with total power consumption limited to 1600 W.

The proposed system addresses critical gaps in the design and optimization of multifunctional HVAC devices, offering a scalable and adaptable solution for diverse applications in residential, commercial, and industrial settings. Future efforts to validate the system through empirical testing and refine its manufacturing feasibility will further solidify its position as a sustainable, high-performance solution for indoor environmental control. The integration of bionic design and advanced optimization technologies in this study represents a significant step forward in the development of next-generation HVAC systems.

## Figures and Tables

**Figure 1 biomimetics-10-00159-f001:**
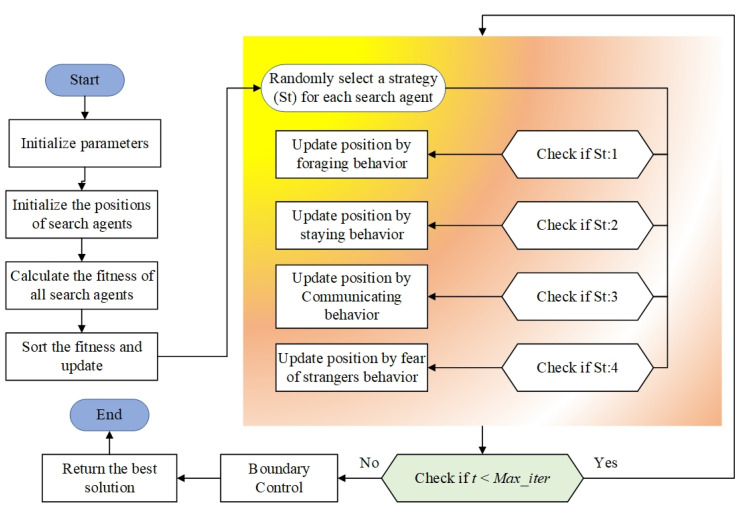
Flowchart of the PO algorithm.

**Figure 2 biomimetics-10-00159-f002:**
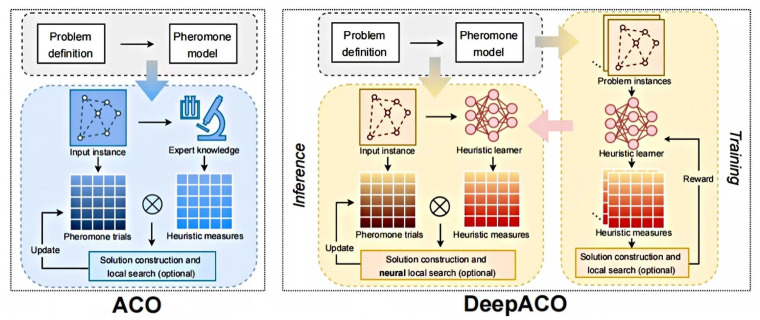
Flowchart of the DeepACO algorithm.

**Figure 3 biomimetics-10-00159-f003:**
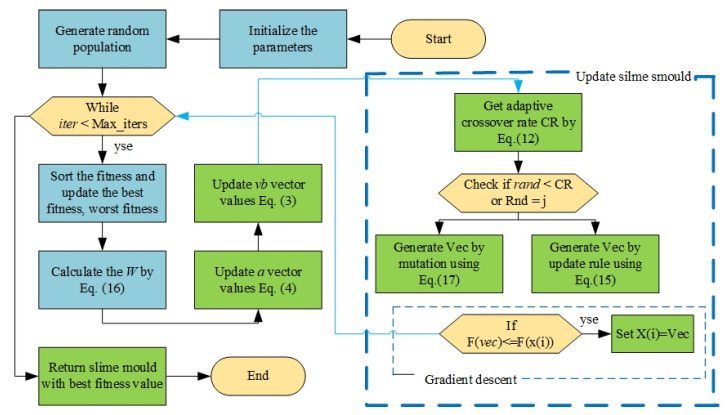
Flowchart of the Slime Mold Algorithm.

**Figure 4 biomimetics-10-00159-f004:**
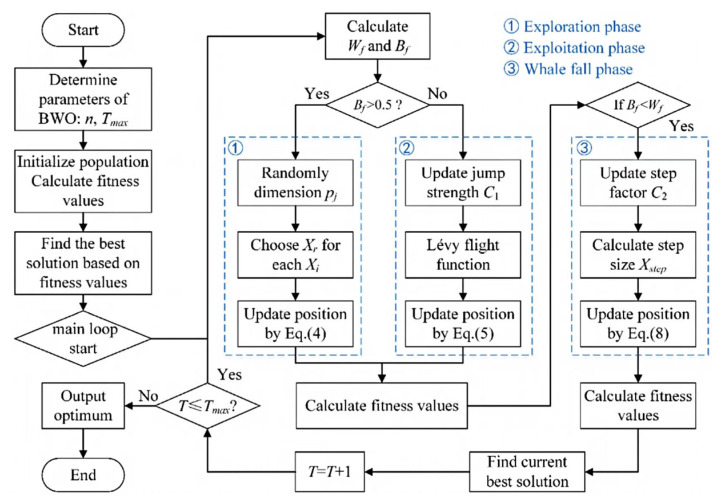
Flowchart of the BWO algorithm.

**Figure 5 biomimetics-10-00159-f005:**
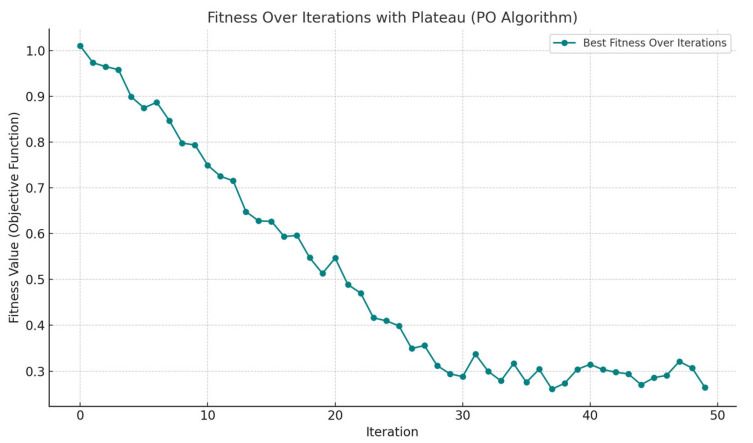
Convergence of the PO algorithm.

**Figure 6 biomimetics-10-00159-f006:**
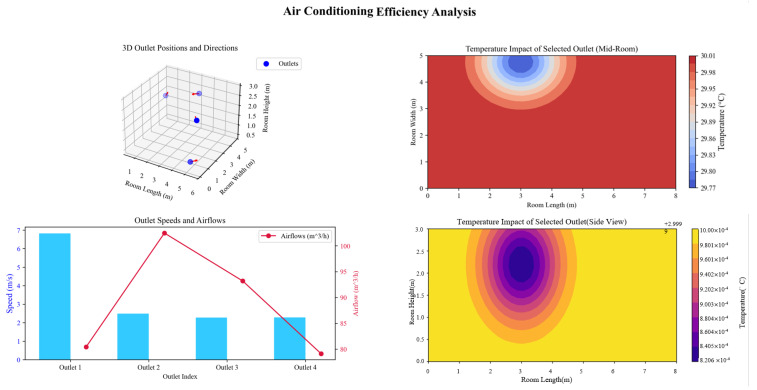
Impact of outlet parameters on air conditioning efficiency.

**Figure 7 biomimetics-10-00159-f007:**
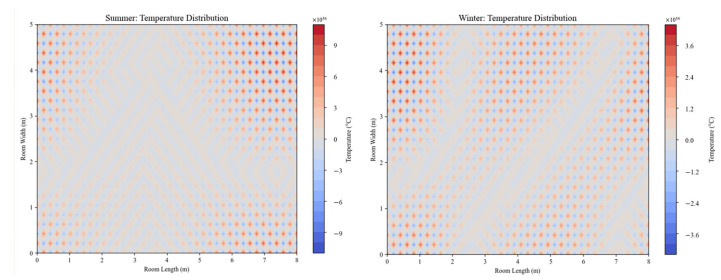
Indoor temperature dynamics: summer vs. winter simulation.

**Figure 8 biomimetics-10-00159-f008:**
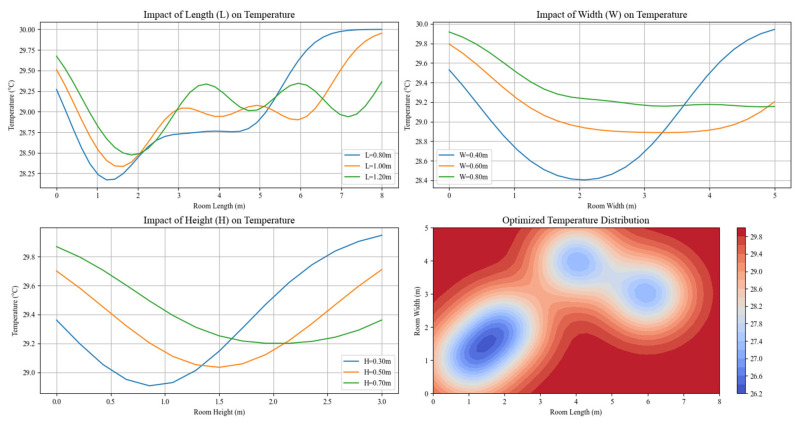
Impact of air conditioning dimensions on temperature distribution.

**Figure 9 biomimetics-10-00159-f009:**
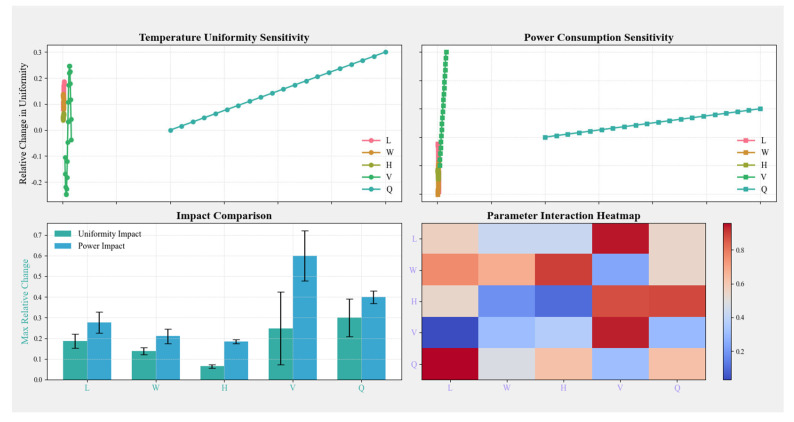
Sensitivity analysis for air conditioning partial modeling.

**Figure 10 biomimetics-10-00159-f010:**
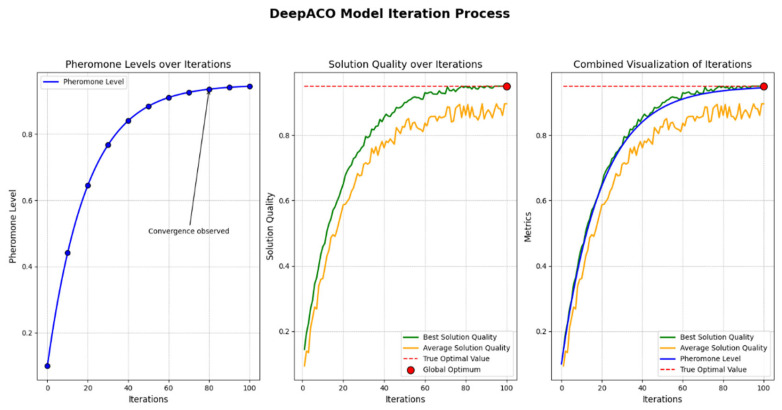
DeepACO model iteration process.

**Figure 11 biomimetics-10-00159-f011:**
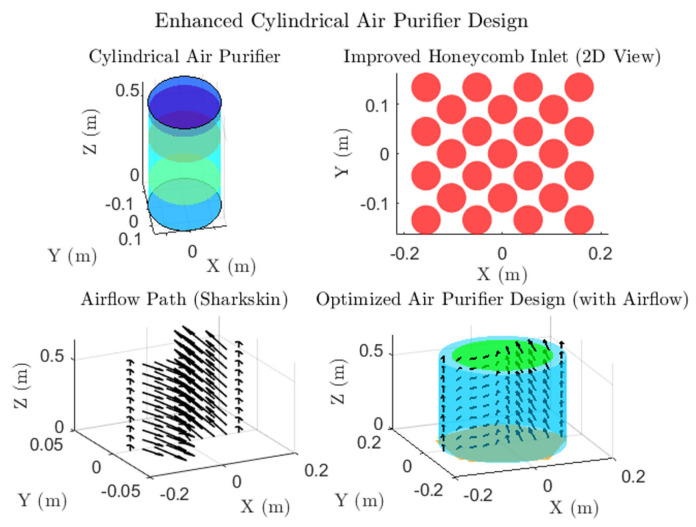
Air purifier design drawing.

**Figure 12 biomimetics-10-00159-f012:**
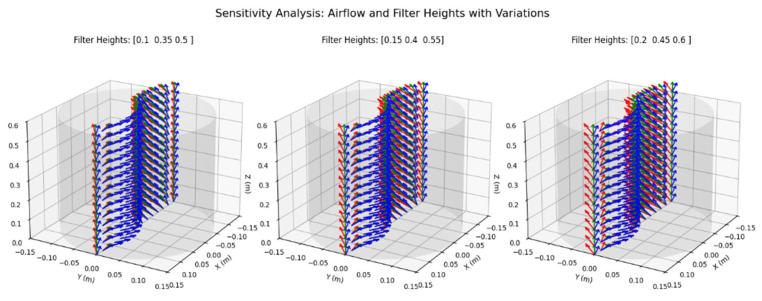
Sensitivity analysis for air purifier partial modeling.

**Figure 13 biomimetics-10-00159-f013:**
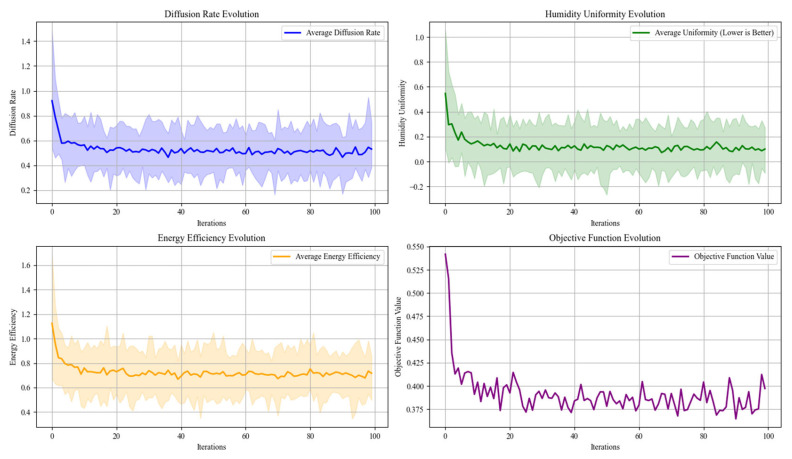
SMA iteration process.

**Figure 14 biomimetics-10-00159-f014:**
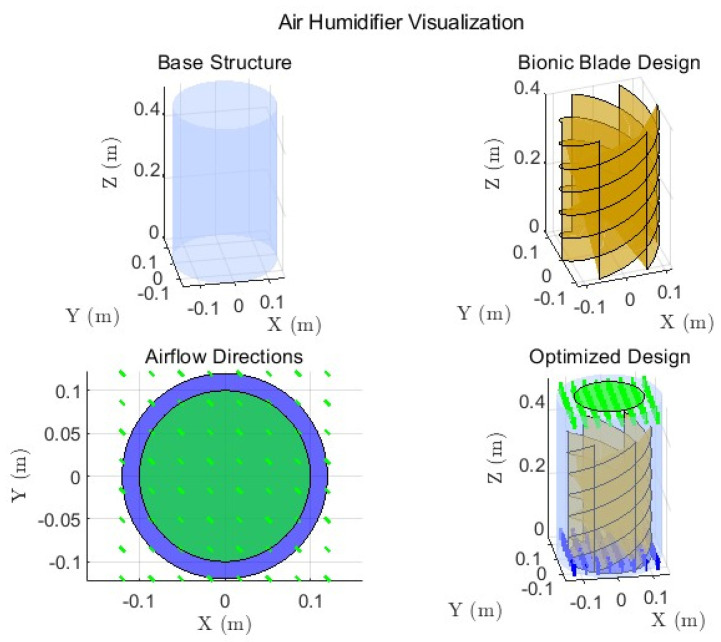
Air humidifier visualization.

**Figure 15 biomimetics-10-00159-f015:**
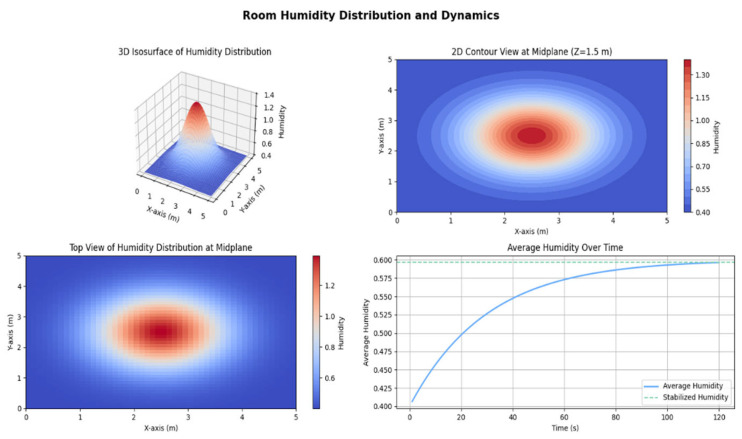
Room humidity distribution and dynamics.

**Figure 16 biomimetics-10-00159-f016:**
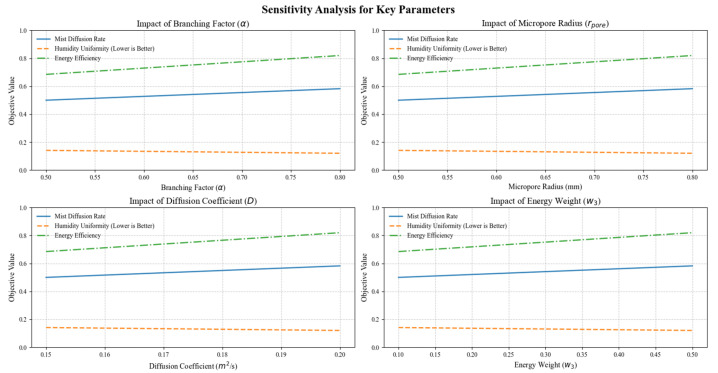
Sensitivity analysis for humidifier partial modeling.

**Figure 17 biomimetics-10-00159-f017:**
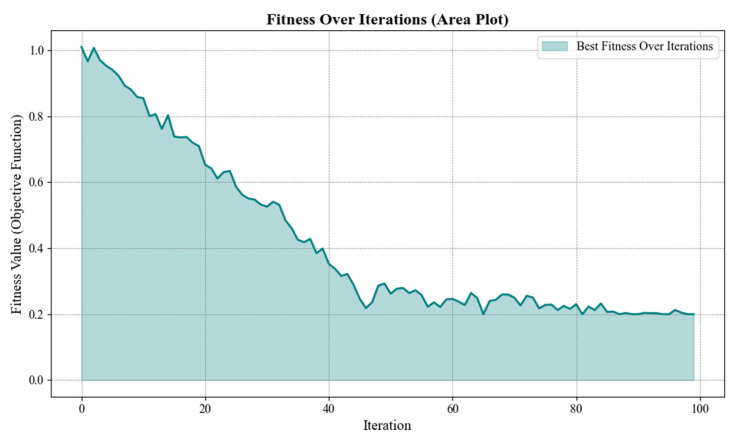
BWO algorithm iteration process.

**Figure 18 biomimetics-10-00159-f018:**
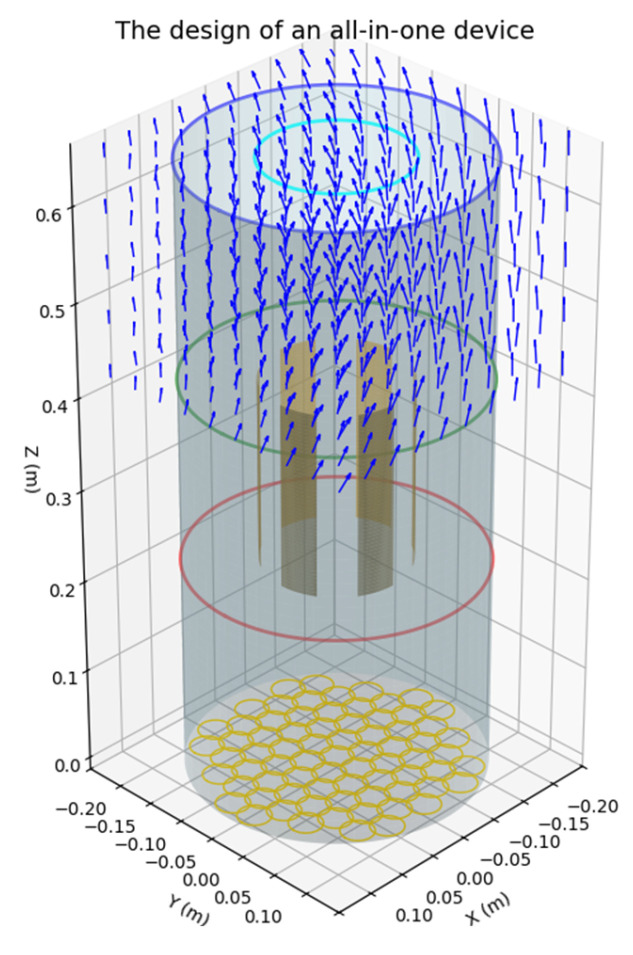
The design of an all-in-one device.

**Figure 19 biomimetics-10-00159-f019:**
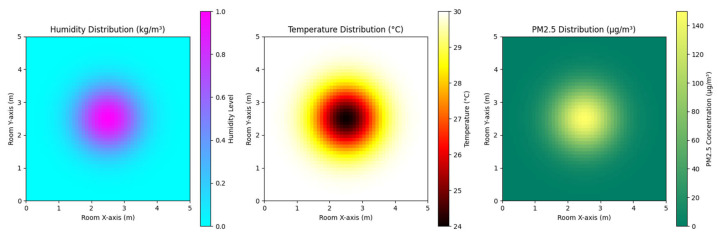
The effect visualization of the all-in-one device.

**Figure 20 biomimetics-10-00159-f020:**
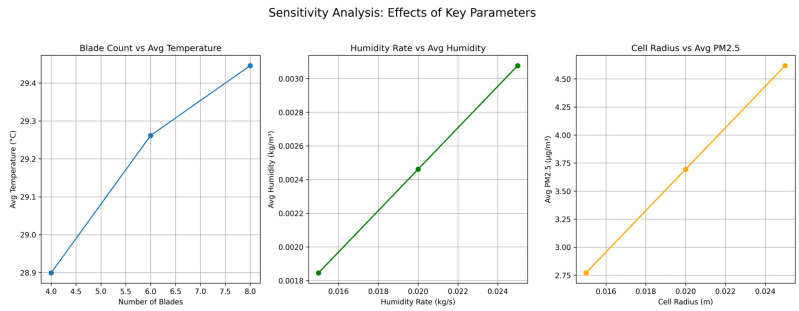
Sensitivity analysis for tri-unit air conditioner modeling.

**Table 1 biomimetics-10-00159-t001:** Summary of combined techniques.

Technique	Advantages	Disadvantages
Energy Simulation and Passive Cooling	Improves energy efficiency, reduces cooling demand	Climate-specific, requires design adjustments
Machine Learning and Computational Optimization	Dynamic optimization, real-time energy savings	High computational demand, complex models
Air Filtration and PCO	Improves IAQ, efficient pollutant removal	Scalability issues, byproduct formation
Humidity Control with Desiccant Wheels	Enhances comfort, energy-efficient humidity control	Performance varies with conditions
Multifunctional HVAC Systems	Space and energy savings, combines multiple functions	Integration and performance challenges
Optimization Algorithms	Solves multi-objective problems, improves efficiency	Computationally intensive, requires expertise
Bionic Design Principles	Enhances airflow, reduces energy use	Limited large-scale application

**Table 2 biomimetics-10-00159-t002:** PO algorithm parameters.

Parameter	Value	Unit
Population Size	100	individuals
Maximum Foraging Area Adjustment	0.1	proportional factor
PO Foraging Search Radius	0.2	proportional factor
Number of Iterations	50	iterations

Note: The algorithm iteratively adjusts design parameters, including the length (*L*), width (*W*), height (*H*), outlet velocity (*V*), and airflow rate (*Q*), until convergence is achieved.

**Table 3 biomimetics-10-00159-t003:** DeepACO algorithm parameters.

Parameter Setting	Value
Learning Phase 1: Heuristic Indicator Training	
Learning Rate	0.001
Batch Size	32
Epochs	500
Dropout	0.2
L2 Regularization Coefficient	0.0001
Dataset Size	10,000/samples
Learning Phase 2: Pheromone Trajectory Learning	
Number of Ants	20
Number of Iterations	500
Evaporation Rate	0.1
Heuristic Importance	1
Pheromone Importance	2
Initial Pheromone Value	1

**Table 4 biomimetics-10-00159-t004:** Slime mold algorithm parameters.

Parameter Name	Value
Population Size	30
Maximum Number of Iterations	500
Sensitivity Initial Value	1
Vibration Parameter Initial Value	0.5
Random Disturbance Parameter	0.5
Positive Feedback Adjustment Weight Initial Value	0.7

**Table 5 biomimetics-10-00159-t005:** BWO algorithm parameter.

Parameter Name	Value
Population Size	30
Maximum Number of Iterations	500
Balance Factor Initial Value	2
Step Size Factor	0.01
Initial Probability of Descent	0.1

**Table 6 biomimetics-10-00159-t006:** Parameters for sensitivity analysis of the first question.

Parameter	Symbol	Range
Length/m	L	0.8 to 1.2
Width/m	W	0.4 to 0.8
Height/m	H	0.3 to 0.7
Velocity/m	V	2 to 8
Airflow/m^3^/h	Q	100 to 300

## Data Availability

The raw data supporting the conclusions of this article will be made available by the authors on request.

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
