# Peer review of "A Bionic-Based Multi-Objective Optimization for a Compact HVAC System with Integrated Air Conditioning, Purification, and Humidification"

_biomimetics, 2025, doi:10.3390/biomimetics10030159_

Round 1
Reviewer 1 Report
Comments and Suggestions for Authors
With a clear approach, thorough optimization frameworks, and intricate mathematical models, the study is well-structured. Advanced algorithms and biomimetic designs are creatively used to solve important HVAC system issues including multifunctionality, noise reduction, and energy efficiency.
There are, however, some drawbacks that will make the work better:
- The language in the manuscript requires improvement as there are technical and grammatical errors. For example,: many articles are missing, like “a ” is missing before “total” in line 27 in the abstract
- The discussion on the advantages and disadvantages of the reviewed combined techniques in the review section (section 2) could be summarized in a table. This would enhance clarity and make it easier for readers to compare the techniques.
- The practical issues of creating these intricate geometries are not covered in the study, which could restrict the system's ability to be widely used. In order to streamline production without sacrificing performance, future research should investigate manufacturing processes.
- Adding real-time flexibility through machine learning algorithms or feedback systems based on sensors could make the system better at responding to changing conditions in the world.
- A more in-depth comparison with current HVAC systems that can do more than one thing would improve the presentation of the paper.
- The paper says that there are big improvements in energy efficiency, but it would be helpful to see a more in-depth investigation at how much energy is saved compared to old methods.
- Cost-effectiveness should be added to elaborate the economic aspects of the hybrid proposed methodology
- Improve the quality of the figures.
Reviewer 2 Report
Comments and Suggestions for Authors
The reviewer would like to thank the authors for presenting the manuscript "A Bionic-Based Multi-Objective Optimization for a Compact HVAC System with Integrated Air Conditioning, Purification, and Humidification", which presents a study on one of the biggest problems of HVAC equipment on the market, that is, normally these equipments focus only on satisfying one parameter of indoor comfort and with this study the authors intend to demonstrate that there is room for the development of equipments with increasingly integrated functions. The reviewer asks the authors for the corrections/clarifications presented in the attached document.

Round 2
Reviewer 2 Report
Comments and Suggestions for Authors
The reviewer would like to thank the authors of the manuscript "A Bionic-Based Multi-Objective Optimization for a Compact HVAC System with Integrated Air Conditioning, Purification, and Humidification" for clarifying the comments and introducing suggested improvements. The reviewer will suggest to the editors the publication of the current manuscript.